# High-Pass Noise Suppression in the Mosquito Auditory System

**DOI:** 10.3390/insects16080840

**Published:** 2025-08-14

**Authors:** Dmitry N. Lapshin, Dmitry D. Vorontsov

**Affiliations:** 1Institute for Information Transmission Problems of the Russian Academy of Sciences (Kharkevich Institute), Bolshoy Karetny per. 19, Moscow 127994, Russia; lapshin@iitp.ru; 2Koltzov Institute of Developmental Biology of the Russian Academy of Sciences, Vavilova 26, Moscow 119334, Russia

**Keywords:** mosquito, *Culex pipiens*, hearing, noise suppression, high-pass filter, phase shift, cutoff frequency, Johnston’s organ

## Abstract

Mosquitoes use their antennae to hear, but their sense of hearing is susceptible to low-frequency background noise from their own flight and the environment. The goal of this study was to understand how mosquitoes overcome this issue. We examined this question using *Culex pipiens* mosquitoes, measuring how their hearing cells respond to different sound frequencies. According to our findings, mosquitoes likely use built-in high-pass filters to block out noise before it reaches their hearing cells. Our modeling calculations, based on experimental data, showed that both male and female mosquitoes filter out low-frequency sounds. However, males do it more effectively, blocking approximately 32 decibels of noise at very low frequencies, compared to 21 decibels in females. This is likely because males need to hear the specific high-pitched buzz of a female’s wings to find a mate. Females, on the other hand, have a more varied hearing ability, which may help them to find a host. These findings help us better understand how mosquitoes hear and why their hearing is so effective, which could be useful for developing new ways to control mosquito populations by disrupting their ability to find mates.

## 1. Introduction

Mosquitoes detect sound using their feathery antennae. When exposed to sound waves, distal segments of the antenna vibrate in proportion to the velocity of air particle displacement [1]. These vibrations are transmitted to the Johnston’s organ (JO) located in the pedicel, a modified second segment of the antenna. The JO contains several thousand primary mechanosensory neurons (PSNs) arranged radially [2], which convert mechanical vibrations into electrical signals.

Similarly to other animals, the auditory perception of mosquitoes is limited by the noise that affects their auditory organs. For a flying mosquito, the sources of noise can be divided into two categories: external, independent of the mosquito’s own activity, and internal, directly related to its behavior. The first group includes convective noise and thermal noise [3,4,5], both of which should act on the mosquito antenna, an organ sensitive to the air particle velocity, mostly in the low-frequency range. Irregular low-frequency disturbances may be treated as noise, decreasing the signal-to-noise ratio of the mosquito auditory system. The second group includes vibrational noise synchronized with wing beats and low-frequency disturbances associated with flight and maneuvering. The influence of wingbeat vibrations on mosquito hearing has been studied in detail [6,7,8,9,10,11], the general conclusion being that they increase rather than decrease the sensitivity to certain frequencies on the basis of nonlinear distortions in the JO.

The efficiency of mechanoreception in PSNs critically depends on keeping the neurons at their operating points, defined as the condition in which their sensitivity is maximized [12], avoiding excessive stretching or compression of the structures performing mechanoelectrical transduction. This is of particular importance for the function of the active amplification mechanism, which involves a complex of dynein–tubulin molecular motors and membrane mechanosensitive ion channels located in the dendrites of the JO PSNs. This active mechanism significantly amplifies sensitivity to very faint signals [13,14] (Figure 1A). If the sensory system deviates too far from its operating point, active amplification becomes either ineffective or entirely disabled. Consequently, the unimpeded transmission of high-amplitude, low-frequency noise from the antenna to PSNs can hamper the system’s ability to maintain high sensitivity. This suggests that the mosquito auditory system must incorporate mechanisms that reduce the impact of such noise on PSNs.

It is well known that many types of sensory receptors act as high-pass filters (HPFs), suppressing stationary or relatively low-frequency signals [15]. Such receptors are often referred to as rapidly adapting, velocity-sensitive, dynamic, or phasic mechanoreceptors. A classic example is the Pacinian corpuscle, where rapid adaptation is facilitated by the redistribution of viscous fluid between layers of connective tissue within the corpuscle’s capsule [16].

One possible solution to the problem of low-frequency noise suppression is the presence of one or more HPFs (Figure 1B) along the signal transmission pathway from the antenna to the sensory neurons. These filters can be implemented through mechanical principles or during subsequent electrical signal processing. The design of HPFs involves a trade-off between minimizing their influence within the functionally important frequency range and maximizing suppression of out-of-band noise.

While the hypothetic HPFs in the mosquito’s peripheral auditory system cannot be accessed directly, their presence and properties may be inferred indirectly by analyzing the phase shift in electrophysiological responses relative to the acoustic stimuli that elicit them. Under continuous sinusoidal stimulation, the output phase of an HPF leads the input phase in steady-state conditions. Theoretically, the phase shift (i.e., the difference between the output and input phases of a sinusoidal signal) introduced by an HPF increases in magnitude as frequency decreases, approaching –90° at frequencies approaching a few Hertz. As frequency increases, the magnitude of the phase shift gradually decreases, asymptotically approaching zero.

The objective of this study was to identify the signs of high-pass filtering in the peripheral auditory system of mosquitoes and to make preliminary estimates of its physical parameters. To this end, we initially implemented a model comprising an HPF and a delay line arranged in series. A comparison of the modeling results with the experimentally obtained data demanded revising the model by adding a second HPF into the signal chain to achieve consistency with the experimental data.

## 2. Methods

Males and females of *Culex pipiens pipiens* L. were captured in the wild near the Oka River (54°51′44″ N, 38°21′28″ E) in the Moscow region of the Russian Federation. Electrophysiological experiments were performed under laboratory conditions at an air temperature of 19–21 °C at the Kropotovo biological station. A total of 34 experiments were conducted with male and 34 with female mosquitoes.

Each experiment consisted of several sequential stages:Searching for a high-quality recording of the sensory receptor activity in the antennal nerve;Determining the optimal direction of sensitivity in the recording (by rotating the vector of the acoustic wave around the axis of the antenna);Measuring the frequency–threshold characteristic (audiogram);Measuring the phase–frequency characteristic.

Individual mosquitoes were fixed by attachment to a 10 × 5 mm copper-covered triangular plate by a flour paste with addition of sodium chloride. Focal extracellular recordings from the axons of the antennal nerve were made with glass microelectrodes (1B100F–4, WPI Inc., Sarasota, FL, USA) filled with 0.15 M sodium chloride and inserted at the scape–pedicel joint. After the penetration of the cuticle, the electrodes had a resistance of 10–60 MΩ.

Neuronal responses were amplified using a home-made AC amplifier (bandpass 5–5000 Hz). For stimulation, two orthogonally oriented stationary speakers were used; they created a vector superposition of acoustic waves at the point of mosquito antennae, as described in detail in [17]. The mosquito was positioned at the crossing of the axes of two speakers in such a way that the antenna’s flagellum was perpendicular to the directions of sound waves originating from each of the two speakers. This approach enabled us to set the desired direction of the acoustic vector relative to the antenna flagellum.

A differential microphone (NR-231-58-000, Knowles Electronics, Itasca, IL, USA) positioned next to the mosquito on a micropositioner with an axial rotation feature recorded the stimulation signals. Neuronal responses and stimulation signals were digitized using an E14-440 A/D board (L-Card, Moscow, Russia) at a 20 kHz sampling rate, with LGraph2 software, version 2.34.

Calibration of the stimulating equipment was performed using the same differential microphone. The differential microphone together with its amplifier was previously calibrated in the far field using a B&K 2253 sound level meter with a B&K 4176 microphone (Brüel & Kjær, Nærum, Denmark). All sound level data in this study are given on a logarithmic scale in dB RMS SPVL (root mean square sound particle velocity level), with a reference level of 0 dB being equal to 4.85 × 10^−5^ mm/s.

At the beginning of the experiment, as the electrode was gradually advanced into the antennal nerve, the preparation was continuously stimulated with tonal pulses of 200 Hz for male and 100 Hz for female mosquitoes. The amplitude of the tones was 60 dB SVPL, duration 80 ms, and period 600 ms, with a smooth rise and fall. In this searching procedure, groups of JO neurons situated orthogonal to the antenna oscillation could be overlooked. To avoid this, the vector of the acoustic wave was periodically changed by 90°.

A response amplitude of 500 µV (peak-to-peak) or more was considered sufficient for subsequent measurements. More detailed methodology for measuring the auditory receptor thresholds was described earlier [10,11,18].

The phasic properties of the auditory response were measured by stimulating the preparation with tonal pulses that incrementally varied from low to high frequencies: 50 dB SPVL and 10 Hz increments for male mosquitoes, and 60 dB SPVL and 5 Hz increments for female mosquitoes; different amplitudes of stimulation reflected the lower auditory sensitivity of female mosquitoes compared to the males [17,18]. The duration of individual pulses was typically 3–4 s; however, it could be increased in the presence of occasional spontaneous interference in the neuronal response. The interval between successive pulses was maintained at a constant 0.15 s.

Both the stimulation control circuit (microphone and microphone amplifier) and the electrophysiological amplification circuit were pre-calibrated for phase shifts, and the data recorded from the neurons were adjusted accordingly.

### Data Processing

Before the measurement of the phase shift, signals in both recording channels (acoustic stimulation and neuronal response) were bandpass-filtered (see example in Figure 2) using the Sound Forge 10 PRO software (Sony, Tokyo, Japan). The purpose of this procedure was to isolate the fundamental frequency in each channel while simultaneously suppressing harmonics and noise. The filter was adjusted to the stimulation frequency in each case. To control for artifacts of digital frequency filtering, pre-synthesized sinusoidal signals of 50 and 100 Hz with predetermined phase shifts between the channels (typically –90°, 90°, and 180°) were filtered in the same way. After the filtering procedure, the phase shift between the signals in the two channels remained unchanged.

To measure the phase difference between the stimulus and response signals, instantaneous phase functions for both signals were calculated in Matlab via the Hilbert transform, as was proposed for the analysis of mosquito flight sounds in [19]. The phase shift was measured in the second half of a tonal pulse when the system entered a steady-state regime. For each tonal pulse, the median value of the phase shift was taken to plot the phase–frequency characteristic (PFC). The Matlab script and the bandpass-filtered source data are available through Figshare (https://doi.org/10.6084/m9.figshare.29493200).

Linear regression plots were calculated based on the PFCs using the least squares method:Ph = k F + D_0_(1)
where Ph is the instant phase shift, k is the coefficient characterizing the slope of the regression line relative to the frequency axis, and D_0_ (initial phase shift) is the point of intersection of the regression plot with the vertical axis at F = 0.

## 3. Results

### 3.1. Modeling a System of a High-Pass Filter and a Delay Line Connected in Series

#### 3.1.1. High-Pass Filter

To illustrate the operation of a high-pass filter (HPF), let us consider a system in which oscillations from the antenna are transmitted to the receptor neurons through a fluid medium with a certain viscosity (Figure 1A). In such a system, the output force of the HPF is proportional to the velocity of displacement of mechanical elements in contact with the fluid. This relationship can be expressed through the following equation:E = ηV(2)
where E is the output force of the HPF, η is a coefficient of viscosity, also including the effects introduced by the geometry and the elasticity of the structures in contact with the fluid, and V is the displacement velocity at the input of the filter.

Given that velocity is the first derivative of the displacement function, a system with viscous friction represents a differentiating element. When a sinusoidal signal is applied to the input of such a filter, the output forms the first derivative of a sine function of the same argument, i.e., the product of the angular frequency of the carrier (ω) and a cosine function. Taking into account that ω = 2πF, where F is a carrier frequency, at the time instance t we obtainE = 2πFη cos(2πFt)(3)

According to this formula, the amplitude of oscillations at the output of the filter is proportional to the frequency of the input signal. At very low frequencies, the transfer coefficient is relatively small, but it increases proportionally with frequency. In real physical systems, the transfer coefficient of an HPF is limited from above, and its general dependence on frequency is shown in Figure 1B. In other words, an HPF is characterized by a cutoff frequency, below which the signal attenuation increases as the frequency decreases. Above the cutoff frequency, the filter becomes less effective (Figure 1B). The cutoff frequency is determined, in particular, by the viscosity of the fluid, through which mechanical perturbation is transmitted and the elasticity of the morphological structures at the input and output of the filter (in our model all these properties are represented by a combined coefficient η).

The second part of Equation (3)—the cosine function—can be viewed as the original sine input function, but phase-shifted by –90°. Theoretically, the phase shift at the HPF output approaches –90° at the lowest frequencies (Figure 3A).

#### 3.1.2. Delay Line

All sensory receptor neurons transfer signals with a certain delay relative to the input (response time delay, τ_0_). To simplify the reasoning, in our model all mechanisms leading to the signal delay, both mechanical transmission and the processes in the receptor neurons, are replaced with a delay line (DL). At the output of the DL, the signal is delayed by time τ_0_ relative to the input. It is important to note that the DL does not introduce any additional alterations to the signal.

The phase shift at the DL output (Ph_2_) increases linearly with the frequency of stimulation (Figure 3B). When the period of the stimulating signal equals the delay time τ_0_, the total phase shift between the input and output of the DL reaches 360°. From this equality, the response delay τ_0_ can be determined from the slope of the experimentally measured phase–frequency characteristic (PFC).

At frequencies approaching zero, when the stimulus period is significantly longer than the delay time, the phase shift becomes negligible and approaches zero.

In the model, the parameters that define the DL properties were determined using averaged data on the measured delay time of the electrophysiological response τ, which was calculated from the slope of the experimentally determined PFCs. Given that in a linear system,k = Ph_2_/F
where Ph_2_ is the phase shift introduced by the DL and F is the signal frequency, we obtain the formula for calculating the delay time from the slope of the experimental PFC:τ_0_ = k/360(4)

However, it is important to note that the slope of the PFC, and consequently the calculated delay time τ, could be increased by the additional phase shift introduced by the hypothetical HPF. The magnitude of this shift could only be determined after identifying the parameters of this HPF. Consequently, the delay time discussed in relation to the DL refers to the value obtained from the experimental data, unless otherwise stated.

#### 3.1.3. Combined Phase Shift of the Model

In a first approximation, we assume that a DL and a hypothetical HPF are connected in series on the path from the antenna to the mechanotransduction area of the receptor neuron. The phase shift at the HPF output (Ph_1_, in angular degrees) is determined byPh_1_ = −arctg(1/(2πFT_0_))(5)
where T_0_ is the time constant of the HPF, and F is the signal frequency in Hz.

The HPF cutoff frequency F_0_ is related to the time constant T_0_ byF_0_ = 1/(2πT_0_)(6)

Then,Ph_1_ = −arctg(F_0_/F)(7)

The phase shift introduced by the DL, in angular degrees, is as follows:Ph_2_ = 360Fτ_0_(8)
where τ_0_ is the delay in the DL, in seconds.

The total phase shift in the system isPh = Ph_1_ + Ph_2_(9)

At the lowest frequencies, the phase shift at the HPF output approaches –90°, while the influence of the delay line on the phase characteristic is minimal. In the frequency–phase shift coordinates, the phase characteristic approaches the vertical axis (zero input frequency) at –90° (Figure 3C,D). At the point where the PFC intersects the vertical axis, the Formula (7) does not allow us to calculate the angle due to division by zero in the F_0_/F ratio.

As illustrated in Figure 3, the calculated graphs depict two HPFs with arbitrary selected cutoff frequencies of 16 Hz (Figure 3C) and 99 Hz (Figure 3D). A comparison of these graphs reveals that, at the higher HPF cutoff frequency, the regression line of the phase characteristic intersects the vertical axis closer to –90°. At frequencies several times higher than the cutoff frequency of the HPF, the DL becomes the predominant factor influencing the total phase shift. Considering the linear dependence of the phase shift introduced by the DL on the frequency, the resulting graph at high frequencies approximates a straight line. If this line is extrapolated toward low frequencies, it should intersect with the vertical axis from –90° to 0°.

The negative value of the phase shift at the point of intersection between the regression line and the vertical axis (i.e., the initial phase shift, D_0_) indicates the presence of the HPF in the system and can be used to estimate the HPF’s cutoff frequency.

### 3.2. Measurement of Phase–Frequency Characteristics of Primary Auditory Neurons

The experimental testing of the hypothesis regarding high-frequency filtering in the mosquito auditory pathway poses significant methodological challenges. At low frequencies, the sensitivity of the JO receptor neurons undergoes a sharp decrease (i.e., their response thresholds increase). Consequently, the magnitude of the stimulus must be considerably augmented during the measurement of auditory thresholds. However, this approach has limitations: at high amplitudes, nonlinear distortions emerge within the JO, giving rise to harmonic components that are integer multiples of the fundamental stimulus frequency. These components, which appear within the frequency range of the best auditory sensitivity (Figure 4), elicit neuronal responses to the harmonics rather than to the fundamental frequency of stimulation. This phenomenon imposes constraints on the accuracy of measurements of low-frequency thresholds. Also, at the lowest stimulation frequencies and high amplitudes, analogous nonlinear processes (i.e., the generation of harmonics) emerge in the hardware of the stimulation system, particularly in the acoustic emitters (speakers). In view of the above, there is an area of uncertainty (shaded gray in Figure 4) within which there are currently no reliable methods for measuring frequency–threshold characteristics within the auditory system of mosquitoes.

The measurement of phase shifts in the low-frequency range is practically impossible for the same reasons as threshold measurements; namely, the difficulty of obtaining an electrophysiological response that would include the first harmonic of sufficient amplitude to measure the phase difference between the input and output signals. However, using the results of modeling, it became possible to overcome this “uncertainty region” at low frequencies and to integrate the experimental and calculated data into a unified system. This was performed by using the PFCs experimentally measured at higher frequencies for calculating the linear regression plots till their intersection with the vertical (F = 0) axis.

Based on the difference in frequency ranges of male and female mosquitoes [18,20], here, in electrophysiological experiments, PFCs were measured in the ranges from 100–160 to 300–360 Hz in male mosquitoes (n = 37) and from 50–80 to 180–220 Hz in female mosquitoes (n = 36). Marginal segments of the PFC curves that exhibited a substantial deviation from linearity were excluded from further analysis.

It should be noted that the initial phase shift D_0_ can only be determined up to a multiple of 180° (±180∙n, where n is an integer). This ambiguity arises because the antennal nerve contains axons from receptor neurons that respond in antiphase to the same antennal deflection [17,21,22]. Figure 5 shows examples of the measured PFCs that are shifted relative to each other by approximately 180°, for pairwise comparison. Data were obtained from males (Figure 5A) and females (Figure 5B). Judging by the different slopes of the PFC curves, the receptor neurons differed in their delay times, leading to additional phase shifts under identical stimulation conditions.

Wave-like deviations from a relatively straight line were observed in almost all measured PFCs, sometimes reaching significant amplitudes (indicated by arrows in Figure 5). Such deviations can be attributed to the influence of responses from narrowly tuned receptor neurons on the summary electrophysiological recording [17,22]. When determining the linear regression parameters, these deviations were excluded. During the subsequent analysis, the values of k (slope) and D_0_ (initial phase shift) were calculated for each experiment, and the delay time τ was then determined from the k values according to Formula (4). The distributions of these parameters are shown in Figure 6, separately for experiments with males (Figure 6A,C,E,G) and females (Figure 6B,D,F,H). The level of uncertainty in the estimation of D_0_ was on average σ = 7.3° for males and σ = 5.2° for females; this led to uncertainty in the calculation of the HPF cutoff frequencies of ca. 6 Hz at 50 Hz for males and of ca. 3 Hz at 25 Hz for females (shown in Figure 6C,D as black bars below the frequency axis; 50 and 25 Hz were taken as central frequencies in the distributions; the estimated uncertainty was ca. 20% lower at lower frequencies and ca. 20% higher towards the higher frequencies).

First, it should be noted that the majority of the obtained D_0_ values fall within the range of negative angles (Figure 6A,B). This supports our initial hypothesis that the HPF is present in the mosquito auditory system on the way from the antenna to the recording site in the antennal nerve. However, a substantial group of D_0_ values lies below –90°, which is inconsistent with the parameters of a single HPF initially assumed in our calculations (Figure 3A). The presence of this group of data cannot be explained solely by the scatter of experimental measurements, as each individual D_0_ value was determined from a sequential series of data points forming the PFC, from which the outliers had already been excluded during the preliminary analysis.

The presence of phase shifts D_0_ < −90° can be explained by hypothesizing the existence of at least two HPFs in the signal transmission pathway between the antenna and the auditory receptor axons, potentially based on different physical principles. In such a scenario, the limitation on the maximum negative deviation of the initial phase shift D_0_ should be extended. To test this hypothesis, we re-estimated the total phase shift in the model incorporating two serially connected HPFs.

### 3.3. Mathematical Model of a System Including Two High-Pass Filters and a Delay Line

When two HPFs are connected in series within a functional chain, their transfer coefficients (gains) are multiplied, and the total phase shift is the sum of their individual phase shifts. In this context, it should be noted that the total number of HPFs in the signal chain could potentially exceed two. However, such scenarios cannot be identified using the method employed in this study due to the inherent ±180° uncertainty in estimating the initial phase shift.

Below, for the sake of simplicity, we will first consider a specific case with two HPFs, which have similar cutoff frequencies.

Essentially, another identical HPF is added to the model that already contains a DL and one HPF. Consequently, the maximum phase shift in the low-frequency region doubles, approaching –180° as F→0. The subsequent methodology of analysis remains unchanged.

### 3.4. Revision of Experimental Data in View of a Model with a DL and Two HPFs

Based on the initial values of phase shift (D_0_) and delay time (τ) obtained in each experiment, we calculated (i) the cutoff frequencies of the two serially connected HPFs (F_0_, Figure 6C,D), and (ii) the delay time τ_0_ calculated according to Formula (6) after computationally excluding the phase shift caused by HPFs (Figure 6E,F).

On average, the calculated delay time τ_0_ was 7.1 ms (SD = 1.5 ms) for males (distribution in Figure 6E) and 7.4 ms (SD = 2.8 ms) for females (distribution in Figure 6F). The delay time τ_0_ was 7.5% shorter than the delay time τ measured directly from the slopes of the experimental PFCs. This difference is attributed to the influence of the HPF(s).

To determine the suppression level for a 10 Hz sinusoidal signal, the classic formula for the HPF transfer coefficient (gain) is used:K_1_ = d/√(1 + d^2^)(10)
where d = F/F_0_.

For two serially connected HPFs, their transfer coefficients are multiplied. Given the assumption that the filters have identical parameters,K_12_ = d^2^/(1 + d^2^)(11)

The attenuation level (L) is the reciprocal of the transfer coefficient, expressed in decibels (dB):L = 20 ∙ log_10_ (1/K_12_), dB(12)

In males, the attenuation level for the 10 Hz signal averaged 32 dB (SD = 5 dB) (Figure 6G), while in females, it averaged 21 dB (SD = 11 dB) (Figure 6H).

The data for females includes a group of results with calculated HPF cutoff frequencies clearly below 10 Hz (Figure 6D); this group corresponds to attenuation levels close to 0 dB (Figure 6H). In other words, in four experiments with female mosquito’s auditory neurons, they showed virtually no signs of low-frequency filtering.

The calculated amplitude–frequency characteristics of a single and a paired HPF with 50 Hz cutoff frequencies are shown in Figure 7 with blue and red curves. Although an assumption of equality of the cutoff frequencies in a pair of HPFs does not require introducing any additional parameters, it is likely that biological HPFs in a signal transmission chain possess different parameters. To test how non-identical HPFs would behave in the framework of our model, we briefly examined a pair of HPFs, one having a cutoff frequency of 33 Hz, which is 1.5 times lower than 50 Hz. As the total phase shift in the system was based on experimental data and, therefore, did not depend on subsequent interpretations, we introduced a condition that the total phase shift for the standard test signal (50 Hz) should retain the same value. In this case, the cutoff frequency of the second HPF is a dependent value and can be calculated using Formula (7). Under these assumptions, we evaluated the level of signal attenuation for each of the HPFs alone and for the pairs of equal and different HPFs (Figure 7).

For the first set of unequal HPFs with F_1_ = 33 Hz (50/1.5) and F_2_ = 75 Hz, the phase shifts were Ph_1_ = −34° and Ph_2_ = −56° (at 50 Hz). The attenuation level at 10 Hz (vertical dotted line in Figure 7) was 28.4 dB, which was almost equal to that of two similar HPFs (28.3 dB).

## 4. Discussion

According to electrophysiological and behavioral experiments, male mosquitoes possess extremely high auditory sensitivity, with response thresholds comparable to those of mammals [5,23]. At the same time, such sensitivity is achieved under conditions of high levels of noise from various sources. In particular, at a typical flight speed of 0.4–0.6 m/s during swarming [23], the oncoming uneven airflow that deflects the mosquito’s antennae exceeds the measured auditory thresholds by a factor of a million, or by ca. 120 dB [24]. It is difficult to imagine a single mechanism capable of ensuring such a high degree of noise immunity. It is more likely that several mechanisms based on different principles are involved at successive stages of perception and processing of acoustic information in the mosquito auditory system, collectively providing the required level of noise filtering.

One such mechanism may be a HPF—or several HPFs—located within the peripheral part of the auditory system.

A comparison of attenuation levels in systems with one and with two HPFs showed that in the latter case, the suppression of low-frequency signals was significantly more effective (Figure 7). Thus, double filtering provides clear advantages in ensuring the noise immunity of the mosquito’s JO. This is especially important for the auditory system of male mosquitoes, whose high auditory sensitivity is crucial for their reproductive success.

The introduction of a more than two-fold (75/33) difference between the cutoff frequencies in the pair of HPFs has virtually no effect on the resulting noise suppression efficiency (Figure 7). However, an increase in the cutoff frequency of an HPF to 100 Hz and higher leads to decrease in sensitivity in the frequency range of mosquito acoustic perception. This is already notable at the 75 Hz cutoff frequency: the lower-frequency part of the distribution of neuronal tuning frequencies in *Culex* male mosquitoes [20] is shown in Figure 7 for comparison.

Theoretically expected suppression level (ca. 0.04, or 28 dB at 10 Hz for a double HPF with a 50 Hz cutoff frequency, Figure 7) indicates that such filters suppress a relatively small portion of the low-frequency interference acting on the input of the mosquito auditory system (ca. 120 dB). Nevertheless, their contribution is important, as it reduces the likelihood of overloading the auditory input by the oncoming airflow or gravity on the mosquito’s antennae.

The physical implementation of HPFs in the mosquito auditory system remains speculative. The need to keep the PSNs at their operating points implies that the HPF, to be effective, must be located prior to the mechanotransduction process. One of the possible mechanisms would be signal transmission through the liquid, either extra- or intracellularly, before transduction into the receptor potential. Also, an HPF does not necessarily have to be incorporated into the mechanical part of the signal processing chain. A similar function may be performed by a complex of ionic mechanisms that stabilize the membrane potential of receptor neurons or other mechanisms responsible for adaptation to slowly changing stimuli.

Given the relatively wide distribution of individual cutoff frequencies (Figure 6C,D), each auditory PSN must contain an HPF located directly at its input. Individual HPFs allow operating points to be maintained, and thus high sensitivity, for a large number of sensory neurons despite their different orientations relative to the antennal flagellum. Accordingly, the most likely location of one of the structures implementing high-pass filtering is in the distal part of the dendrite of the auditory neuron within the A1-type sensilla of the JO [1,2].

Our results indicate that characteristics of HPFs are different in male and female mosquitoes. In addition, the auditory system of females includes a group of low-frequency-tuned PSNs without notable signs of high-pass filtering (Figure 6D,H). These findings are consistent with earlier experiments on females, where no decrease in sensitivity was observed in several PSNs tuned to the lowest frequencies [22]. However, the frequency characteristics of such neurons can apparently be significantly modified under the influence of neuromodulaion by, for example, octopamine [25]. In *Anopheles gambiae*, octopamine shifts up the mechanical tuning frequency by increasing the stiffness of the flagellum, and it controls the erection of antennal fibrillae in males [26]. In general, context-dependent neuromodulation may be another mechanism to increase immunity to low-frequency noise by tuning the auditory system to higher frequencies.

There is growing evidence that the female auditory system plays a role in both reproductive behavior and the localization of potential hosts by detecting movement or vocalizations [27,28,29]. The capacity to perform such a wide array of tasks has, in turn, given rise to greater diversity in the characteristics of the JO PSNs. As evidenced by the experimental findings of this study, this diversity manifests in a widened distribution of the HPF cutoff frequencies (Figure 6C,D), response delay times (latencies) (Figure 6E,F), and suppression levels of low-frequency signals (Figure 6G,H) in female mosquitoes compared to males.

In the data obtained from females, the main group of the HPF cutoff frequencies is concentrated in the low-frequency range (Figure 6D), in contrast to the distribution observed in males (Figure 6C). This difference is consistent with the data on the neuronal frequency tuning in female [18,22] and male mosquitoes [17]. Also, female mosquitoes possess a subgroup of sensory neurons with negligible frequency filtering (Figure 6D,H; note the values concentrated close to zero in both diagrams).

## Figures and Tables

**Figure 1 insects-16-00840-f001:**
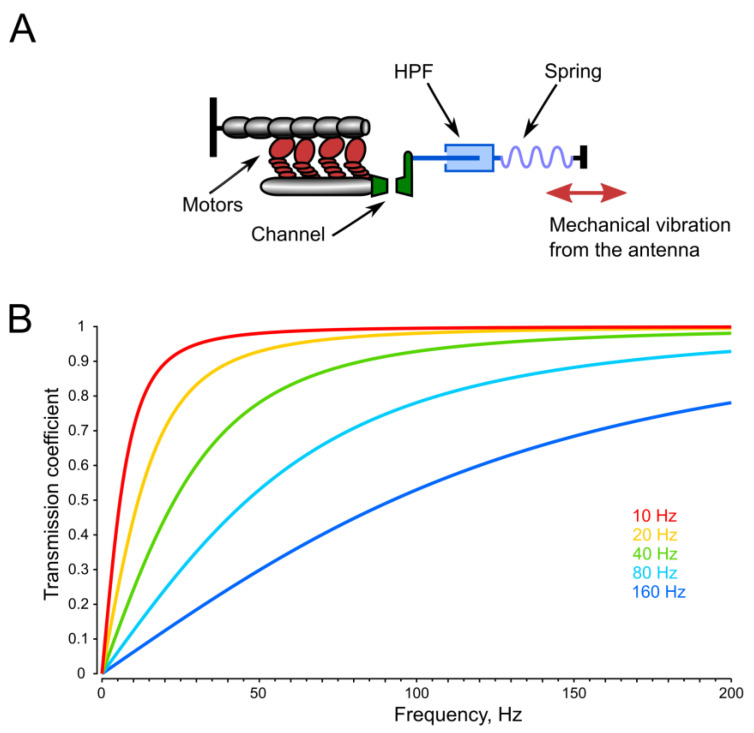
Integration of a high-pass filter into the mechanical transmission pathway from the antenna to the active amplification mechanism of the sensory neuron. (**A**)—Simplified diagram of the mechanotransduction module, consisting of a single ion channel, an adaptive molecular motor, and an elastic element (adapted from [14]). A high-pass filter (HPF) is added as an additional component in the signal transmission chain. In this model, force is transmitted through a liquid medium; due to viscous properties of liquid, the force at the output of the filter (towards the neuronal membrane with the ion channels) is proportional to the velocity of displacement at the filter input. (**B**)—Calculated signal transmission curves of HPFs with different cutoff frequencies from 10 to 160 Hz.

**Figure 2 insects-16-00840-f002:**
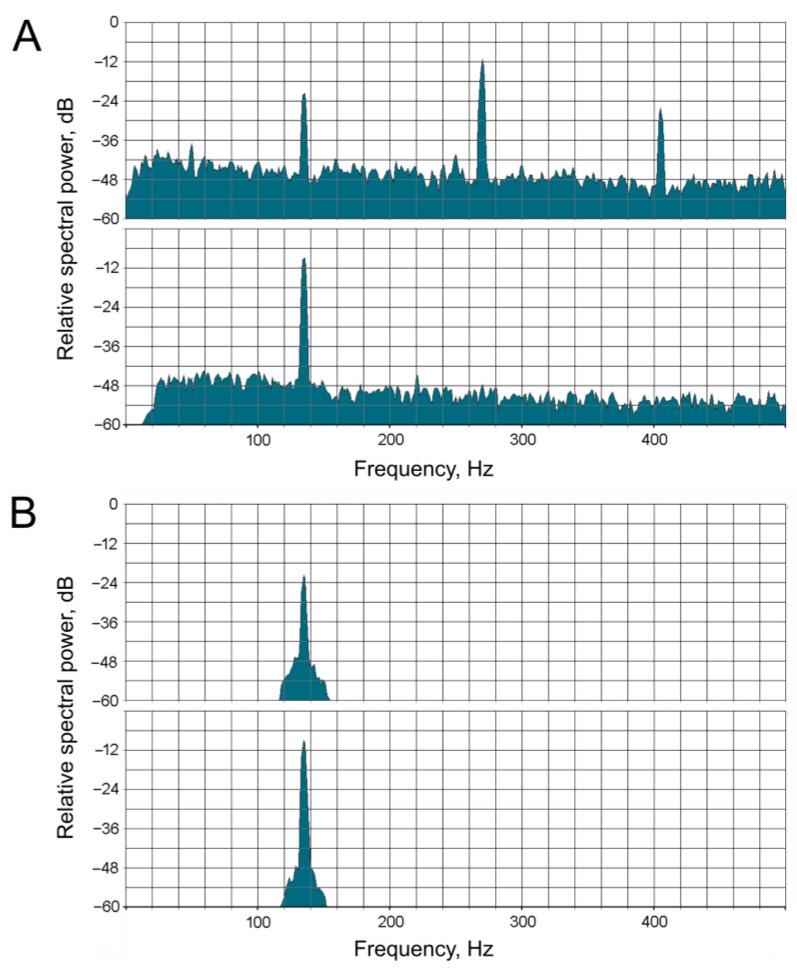
Example of frequency-domain processing of the PSN’s responses in a *Culex pipiens pipiens* female mosquito. (**A**)—Spectra of the original electrophysiological response (upper spectrogram) and the acoustic stimulus recorded by the microphone (lower spectrogram). (**B**)—The same spectra after digital frequency filtering of the signals, prepared for phase measurement. The filling frequency of the sinusoidal acoustic stimulus was 135 Hz.

**Figure 3 insects-16-00840-f003:**
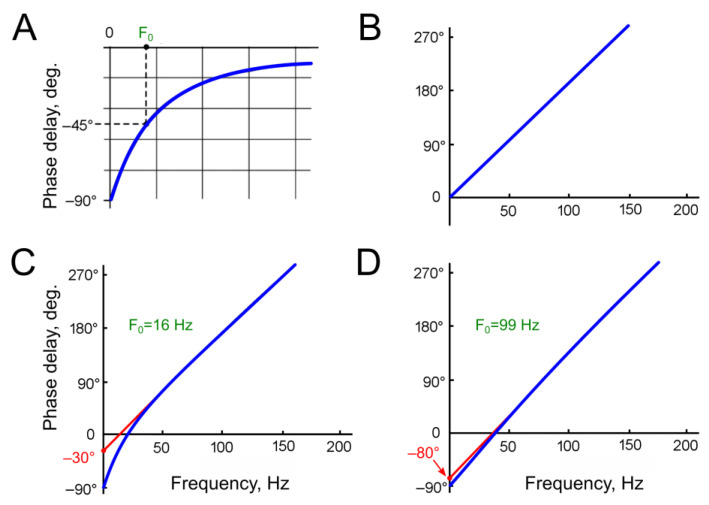
Examples of model phase–frequency characteristics (PFCs, phase shift as a function of frequency). (**A**)—General form of the phase–frequency characteristic (PFC) of a first-order high-pass filter (HPF). (**B**)—PFC of a delay line (DL, τ_0_ = 7.8 ms). (**C**)—Resulting phase–frequency characteristic of a HPF with cutoff frequency F_0_ = 16 Hz and τ_0_ = 7.8 ms. (**D**)—Resulting phase–frequency characteristic of a HPF with cutoff frequency F_0_ = 99 Hz and τ_0_ = 7.8 ms. The regression lines, extrapolated from the phase–frequency plots, intersect the vertical axes at –30° (**C**) and –80° (**D**). On experimental plots, the position of this intersection on the vertical axis (initial phase shift, D_0_) reflects the degree of the HPF contribution to the resulting PFC.

**Figure 4 insects-16-00840-f004:**
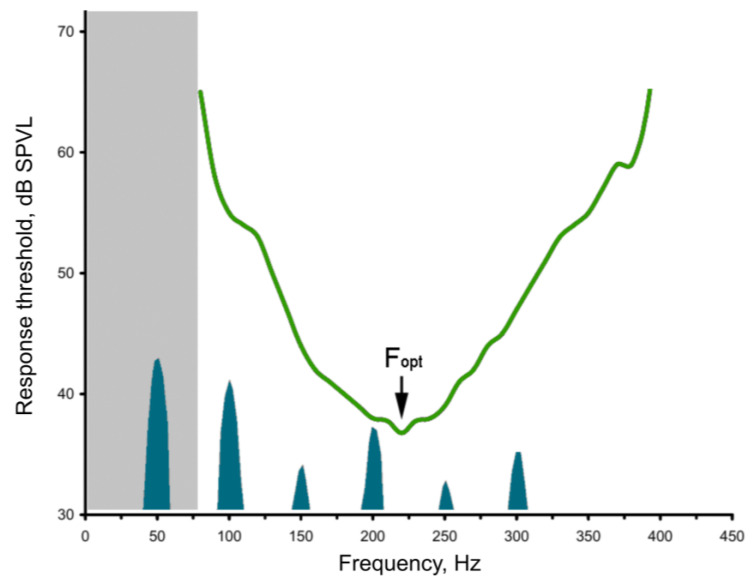
Example of an audiogram of a JO PSN of a male mosquito above the frequency spectrum of harmonic series formed from the 50 Hz sinusoidal signal as a result of nonlinear distortion. Spectral peaks of harmonics are shown below the audiogram curve; some of them fall within the range of maximum auditory sensitivity. The gray-shaded area on the left indicates the frequency range where the auditory thresholds are difficult to measure.

**Figure 5 insects-16-00840-f005:**
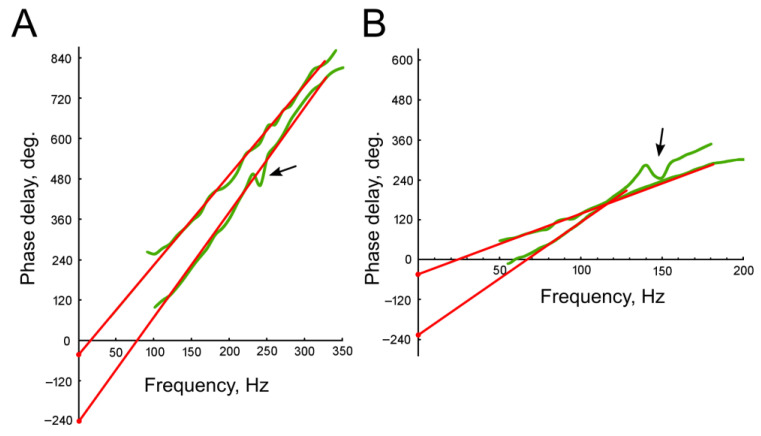
Examples of individual phase–frequency characteristics (PFCs) measured in two male (**A**) and two female (**B**) mosquitoes. Experimentally measured plots are shown in green. The regression plots, extrapolated from the corresponding PFCs, are shown in red. Arrows indicate the regions of increased irregularity of phase shift, which were excluded from subsequent regression analysis. The intersection of the regression plots with the vertical axis reveals the initial phase shift values (D_0_).

**Figure 6 insects-16-00840-f006:**
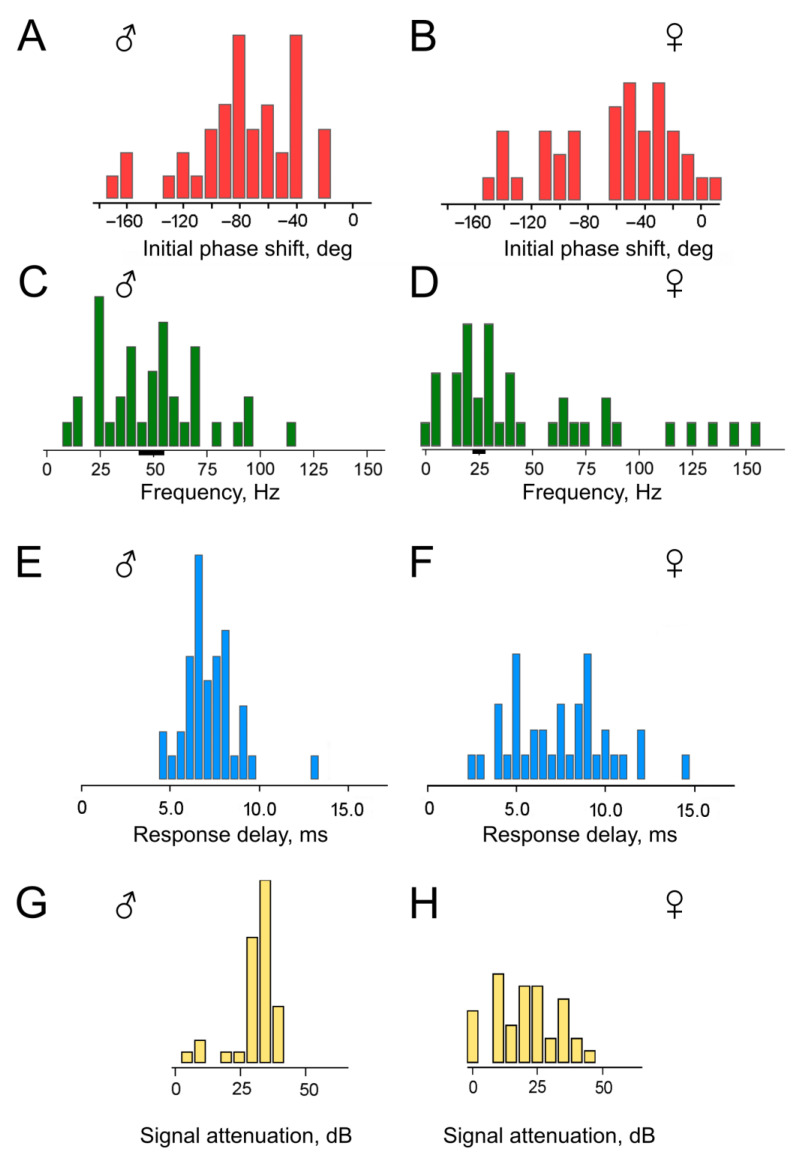
Distributions of parameters calculated on the basis of the experimental data on *Culex pipiens pipiens* mosquitoes. (**A**,**B**)—Initial phase shift values, D_0_ (bin width of the histograms is 10°). (**C**,**D**)—HPF cutoff frequency values, F_0_ (bin width of the histograms is 5 Hz; ranges of uncertainty of estimates are shown as black bars at 50 and 25 Hz in (**C**) and (**D**), respectively). (**E**,**F**)—Delay time values, τ_0_ (bin width of the histograms is 1 ms); (**G**,**H**)—attenuation levels of a 10 Hz sinusoidal signal at the output of a system composed of two high-pass filters with a 50 Hz cutoff frequency (bin width of the histograms is 5 dB). (**A**,**C**,**E**,**G**)—Males; (**B**,**D**,**F**,**H**)—females.

**Figure 7 insects-16-00840-f007:**
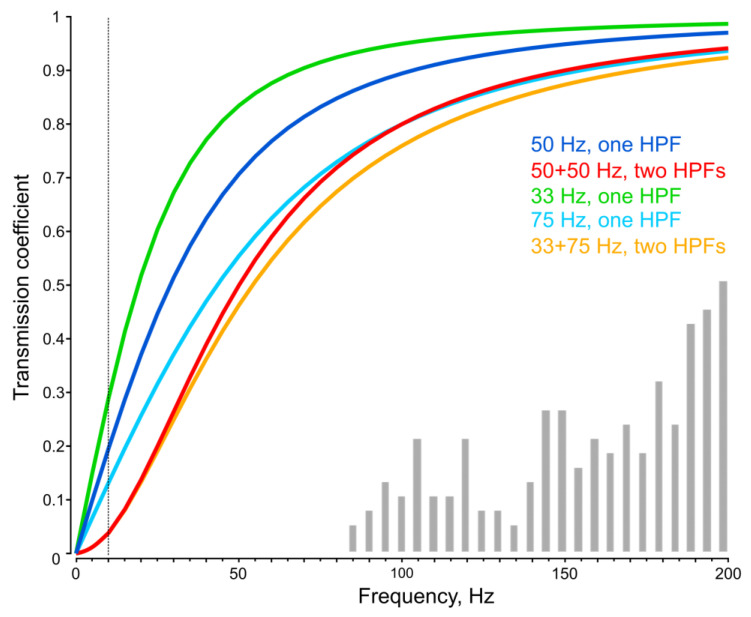
Examples of calculated amplitude–frequency characteristics of single HPFs and pairs of HPFs connected in series with identical cutoff frequencies (50 Hz), typical for male mosquitoes, and with two different cutoff frequencies (33 Hz, 75 Hz). Vertical axis shows the transmission coefficient (K) of the system. Vertical dotted line at 10 Hz shows the levels of attenuation at its intersections with the curves. The level of attenuation at 10 Hz is 14 dB (K_1_ = 0.2) for a single HPF (50 Hz), 28.3 dB (K_12_ = 0.04) for two HPFs (50 + 50 Hz), and 28.4 dB for two different HPFs (33 + 75 Hz). Low-frequency part of the distribution of neuronal tuning frequencies measured in *Culex* male mosqutoes [20] is shown in gray bars.

## Data Availability

The original data presented in the study are openly available from FigShare at https://doi.org/10.6084/m9.figshare.29493200.

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
