# Peer review of "High-Pass Noise Suppression in the Mosquito Auditory System"

_insects, 2025, doi:10.3390/insects16080840_

Round 1
Reviewer 1 Report
Comments and Suggestions for Authors
The paper “High-Pass Noise Suppression in the Mosquito Auditory System” is timely, well written and presented logically. It has been demonstrated that the frequency tuning of the male auditory system enables male mosquitoes to detect females within a swarm without their auditory reception being masked by the swarm noise. However, to my understanding, until this paper, potential masking of the mosquito peripheral auditory system by low frequency noise has not been considered. The experiments described in this paper are model lead. This is because of the extreme technical difficulties involved in measuring neural response to low frequency tones in the mosquito peripheral auditory system. The initial model adopted by the authors consists of a high pass filter followed by a delay line. The delay line represents the sensory processing time between detection of the acoustic signal and the neural response being measured and introduces a fixed delay, regardless of the acoustic stimulus frequency. The high pass filter introduces a phase delay that depends on the cut-off frequency of the filter. This model satisfied data collected from female Johnstone organs, but not those of male mosquitoes. Data from male mosquitoes was fitted better by a delay line followed by two high pass filters. The physical nature of the filters, but not their origin was discussed. Discovering their origin will require a different approach from that described in the paper.
A few points for consideration.
Lines 58-59. “If the sensory system deviates….” This implies that that the HPF must be located prior to the mechanotransduction process. I think a statement to this effect makes it clear that the HPF has to operate early on in the auditory process to be effective.
Lines140-141. Would this be clearer if you said “continuously stimulated with tonal pulses of 200 Hz for male and 100 Hz for female mosquitoes. The amplitude of the tones were 60 dB SVPL, duration 80 ms, period 600 ms”. What about onsets and offsets? Were they cosign functions to avoid onset -offset transients?
Line 227. 7.8 ms seems quite a long delay, between stimulus onset and neural response. Is the delay much shorter for the receptor potential? It might have been expected that a high pass filter would have operated prior to the receptor potential and in this situation the delay line would also be shorter. Do you have any views about this? Is your model dependent on where you record the electrophysiological response to the tones?
Line 307 and Fig 4 shaded grey. Rather than using speakers to deliver acoustic signals to the mosquito auditory system, could you avoid the problems associated with the speakers by using electrostatic displacement of the antennae?
Line 315. Measurement of low frequency phase shifts with an AC coupled amplifier can be problematic. Can you use a DC amplifier, perhaps with a recentring circuit?
Figure 5. Are the data presented in this figure from two different male and female mosquitoes, or two different measurements from the same male and female mosquito?
Figure 7. Could the male second HPF have anything to do with the fact that the male mosquito flagella are more plumose than those of females? When the fibrillae of the Flagellum and Johnston's Organ of Anopheles gambiae M-Form Mosquitoes are extended, the frequency tuning shifts upwards and a low frequency peak is lost (Pennetiere et al, 2010).
Reviewer 2 Report
Comments and Suggestions for Authors
This study presents a sophisticated electrophysiological and modeling approach to investigate high-pass filtering (HPF) in the auditory system of Culex pipiens mosquitoes. The work addresses a significant gap in understanding how mosquitoes maintain auditory sensitivity amid low-frequency noise. The dual-HPF model proposed to explain phase shifts >90° is innovative, and the sex-specific differences in filtering efficiency provide valuable insights into auditory adaptations. The manuscript is well-structured, methods are rigorous, and conclusions are largely supported by data. I recommend minor revisions to enhance clarity and contextualize findings.
Key Limitations and Suggestions:
Biological Substrates for HPFs: While the dual-HPF model fits the data, the physical implementation remains speculative. Discuss potential anatomical correlates or cite relevant work (e.g., Pacinian corpuscle analogies, l. 76–79).
Model Assumptions: The assumption of identical cutoff frequencies for both HPFs (l. 379–380) lacks biological justification. Address whether non-identical filters could produce similar phase shifts.
"Uncertainty Region" in Low Frequencies: Clarify how the dual-HPF model overcomes the low-frequency measurement limitations (Fig. 4). Quantify the extrapolation uncertainty in cutoff frequency estimates (e.g., error bounds in Fig. 6C,D).
Phase Ambiguity (±180°): Emphasize how neuron antiphase responses (l. 328–331) impact D₀ interpretation. Could this ambiguity affect HPF cutoff estimates?
Minor Suggestions:
Abstract: Specify that phase shifts >90° necessitated the dual-HPF model.
Introduction: Define "operating point stability" (l. 52–54) more accessibly for non-specialists.
Methods:
Signal Processing: Justify bandpass filter parameters (l. 161–168). How were harmonics avoided during filtering (Fig. 2)?
Stimulus Amplitude: Explain why higher amplitudes were used for females (60 dB SPVL) vs. males (50 dB SPVL) during PFC measurements (l. 150–152).
Delay Line Model: Clarify if τ₀ includes synaptic/axonal delays beyond mechanical transmission (l. 234–239).
Results:
Female Heterogeneity: Highlight the "negligible filtering" subgroup (l. 406–408) in the main text (not just Fig. 6D,H).
Discussion:
Evolutionary Context: Expand on why female PSNs exhibit broader tuning (l. 456–465). Does this relate to host detection?
Noise Immunity: Quantify how 32 dB suppression (males) reduces the 120 dB convective noise challenge (l. 424–434). Is dual-HPF sufficient, or are additional mechanisms needed?
Neuromodulation: Briefly cite octopamine effects (Vorontsov & Lapshin, 2024) earlier when discussing female PSN diversity (l. 449–452).
References:
Format all references per journal guidelines (italicize species names: Culex pipiens).
- 32: "mechanosen- / sory" → "mechanosensory"
- 211: "proportionally" → "proportional"
- 362: "high-frequency" → "high-pass" (context: filtering)
- 478–483: Expand abbreviations (DL, JO, etc.) at first use.
